# Single-Agent Policy Tree Search With Guarantees

**Laurent Orseau**
DeepMind, London, UK
`lorseau@google.com`

**Levi H. S. Lelis**[*]
Universidade Federal de Viçosa, Brazil
`levi.lelis@ufv.br`

**Tor Lattimore**
DeepMind, London, UK
`lattimore@google.com`

**Théophane Weber**
DeepMind, London, UK
`theophane@google.com`

## Abstract

We introduce two novel tree search algorithms that use a policy to guide search. The first algorithm is a best-first *enumeration* that uses a cost function that allows us to prove an *upper bound* on the number of nodes to be expanded before reaching a goal state. We show that this best-first algorithm is particularly well suited for "needle-in-a-haystack" problems. The second algorithm is based on *sampling* and we prove an *upper bound on the expected* number of nodes it expands before reaching a *set* of goal states. We show that this algorithm is better suited for problems where many paths lead to a goal. We validate these tree search algorithms on 1,000 computer-generated levels of Sokoban, where the policy used to guide the search comes from a neural network trained using A3C. Our results show that the policy tree search algorithms we introduce are competitive with a state-of-the-art domain-independent planner that uses heuristic search.

## 1 Introduction

Monte-Carlo tree search (MCTS) algorithms [Coulom, 2007, Browne et al., 2012] have been recently applied with great success to several problems such as Go, Chess, and Shogi [Silver et al., 2016, 2017]. Such algorithms are well adapted to stochastic and adversarial domains, due to their sampling nature and the convergence guarantee to min-max values. However, the sampling procedure used in MCTS algorithms is not well-suited for other kinds of problems [Nakhost, 2013], such as deterministic single-agent problems where the objective is to find any solution at all. In particular, if the reward is very sparse—for example the agent is rewarded only at the end of the task—MCTS algorithms revert to uniform search. In practice such algorithms can be guided by a heuristic but, to the best of our knowledge, no bound is known that depends on the quality of the heuristic. For such cases one may use instead other traditional search approaches such as A* [Hart et al., 1968] and Greedy Best-First Search (GBFS) [Doran and Michie, 1966], which are guided by a heuristic cost function.

In this paper we tackle single-agent problems from the perspective of policy-guided search. One may view policy-guided search as a special kind of heuristic search in which a policy, instead of a heuristic function, is provided as input to the search algorithm. As a policy is a probability distribution over sequences of actions, this allows us to provide theoretical guarantees that cannot be offered by *value* (*e.g.*, reward-based) functions: we can bound the number of node expansions—roughly speaking, the search time—depending on the probability of the sequences of actions that reach the goal. We propose two different algorithms with different strengths and weaknesses. The first algorithm, called LevinTS, is based on Levin search [Levin, 1973] and we derive a strict upper bound on the number of nodes to search before finding the least-cost solution. The second algorithm,

---

[*]This work was carried out while L. H. S. Lelis was at the University of Alberta, Canada.

called LubyTS, is based on the scheduling of Luby et al. [1993] for randomized algorithms and we prove an upper bound on the expected number of nodes to search before reaching any solution while taking advantage of the potential multiplicity of the solutions. LevinTS and LubyTS are the first policy tree search algorithms with such guarantees. Empirical results on the PSPACE-hard domain of Sokoban [Culberson, 1999] show that LubyTS and in particular LevinTS guided by a policy learned with A3C [Mnih et al., 2016] are competitive with a state-of-the-art planner that uses GBFS [Hoffmann and Nebel, 2001]. Although we focus on deterministic environments, LevinTS and LubyTS can be extended to stochastic environments with a known model.

LevinTS and LubyTS bring important research areas closer together. Namely, areas that traditionally rely on heuristic-guided tree search with guarantees such as classical planning and areas devoted to learn control policies such as reinforcement learning. We expect future works to explore closer relations of these areas, such as the use of LevinTS and LubyTS as part of classical planning systems.

## 2 Notation and background

We write $\mathbb{N}_1 = \{1, 2, \ldots\}$. Let $\mathcal{S}$ be a (possibly uncountable) set of states, and let $\mathcal{A}$ be a finite set of actions. The environment starts in an initial state $s_0 \in \mathcal{S}$. During an *interaction step* (or just step) the environment in state $s \in \mathcal{S}$ receives an action $a \in \mathcal{A}$ from the searcher and transitions deterministically according to a transition function $T : \mathcal{S} \times \mathcal{A} \to \mathcal{S}$ to the state $s' = T(s, a)$. The state of the environment after a sequence of actions $a_{1:t}$ is written $T(a_{1:t})$ which is a shorthand for the recursive application of the transition function $T$ from the initial state $s_0$ to each action of $a_{1:t}$, where $a_{1:t}$ is the sequence of actions $a_1, a_2, \ldots a_t$. Let $\mathcal{S}^g \subseteq \mathcal{S}$ be a set of goal states. When the environment transitions to one of the goal states, the problem is solved and the interaction stops. We consider *tree* search algorithms and define the set of nodes in the tree as the set of sequences of actions $\mathcal{N} := \mathcal{A}^* \cup \mathcal{A}^\infty$. The root node $n_0$ is the empty sequence of actions. Hence a sequence of actions $a_{1:t}$ of length $t$ is uniquely identified by a node $n \in \mathcal{N}$ and we define $d_0(n) = d_0(a_{1:t}) := t$ (the usual depth $d(n)$ of the node is recovered with $d(n) = d_0(n) - 1$). Several sequences of actions (hence several nodes) can lead to the same state of the environment, and we write $\mathcal{N}(s) := \{n \in \mathcal{N} : T(n) = s\}$ for the set of nodes with the same state. We define the set of children $\mathcal{C}(n)$ of a node $n \in \mathcal{N}$ as $\mathcal{C}(n) := \{na | a \in \mathcal{A}\}$, where $na$ denotes the sequence of actions $n$ followed by the action $a$. We define the *target set* $\mathcal{N}^g \subseteq \mathcal{N}$ as the set of nodes such that the corresponding states are goal states: $\mathcal{N}^g := \{n : T(n) \in \mathcal{S}^g\}$. The searcher does not know the target set in advance and only recognizes a goal state when the environment transitions to one. If $n_1 = a_{1:t}$ and $n_2 = a_{1:t}a_{t+1:k}$ with $k > t$ then we say that $a_{1:t}$ is a prefix of $a_{1:t}a_{t+1:k}$ and that $n_1$ is an ancestor of $n_2$ (and $n_2$ is a descendant of $n_1$).

A *search tree* $\mathcal{T} \in \mathcal{N}^*$ is a set of sequences of actions (nodes) such that (i) for all nodes $n \in \mathcal{T}$, $\mathcal{T}$ also contains all the ancestors of $n$ and (ii) if $n \in \mathcal{T} \cap \mathcal{N}^g$, then the tree contains no descendant of $n$. The leaves $\mathcal{L}(\mathcal{T})$ of the tree $\mathcal{T}$ are the set of nodes $n \in \mathcal{T}$ such that $\mathcal{T}$ contains no descendant of $n$. A *policy* assigns probabilities to sequences of actions under the constraint that $\pi(n_0) = 1$ and $\forall n \in \mathcal{N}, \pi(n) = \sum_{n' \in \mathcal{C}(n)} \pi(n')$. If $n'$ is a descendant of $n$, we define the conditional probability $\pi(n'|n) := \pi(n')/\pi(n)$. The policy is assumed to be provided as input to the search algorithm.

Let TS be a generic tree search algorithm defined as follows. At any *expansion step* $k \geq 1$, let $\mathcal{V}_k$ be the set of nodes that have been expanded (visited) before (excluding) step $k$, and let the fringe set $\mathcal{F}_k := \bigcup_{n \in \mathcal{V}_k} \mathcal{C}(n) \setminus \mathcal{V}_k$ be the set of not-yet-expanded children of expanded nodes, with $\mathcal{V}_1 := \emptyset$ and $\mathcal{F}_1 := \{n_0\}$. At iteration $k$, the search algorithm TS chooses a node $n_k \in \mathcal{F}_k$ for *expansion*: if $n_k \in \mathcal{N}^g$, then the algorithm terminates with success. Otherwise, $\mathcal{V}_{k+1} := \mathcal{V}_k \cup \{n_k\}$ and the iteration $k + 1$ starts. At any expansion step, the set of expanded nodes is a search tree. Let $n_k$ be the node expanded by TS at step $k$. Then we define the search time $N(\text{TS}, \mathcal{N}^g) := \min_{k>0}\{k : n_k \in \mathcal{N}^g\}$ as the number of node expansions before reaching any node of the target set $\mathcal{N}^g$.

A policy is *Markovian* if the probability of an action depends only on the current state of the environment, that is, for all $n_1$ and $n_2$ with $T(n_1) = T(n_2), \forall a \in \mathcal{A} : \pi(a|n_1) = \pi(a|n_2)$. In this paper we consider both Markovian and non-Markovian policies. For some function cost $: \mathcal{N} \to \mathbb{R}$ over nodes, we define the cost of a state $s$ as $\text{cost}(s) := \min_{n \in \mathcal{N}(s)} \text{cost}(n)$. Then we say that a tree search algorithm with a cost function $\text{cost}(n)$ expands states in *best-first order* if for all states $s_1$ and $s_2$, if $\text{cost}(s_1) < \text{cost}(s_2)$, then $s_1$ is visited before $s_2$. We say that a state is expanded at its *lowest cost* if for all states $s$, the first node $n \in \mathcal{N}(s)$ to be expanded has cost $\text{cost}(n) = \text{cost}(s)$.

| Algorithm 1: Levin tree search. | Algorithm 2: Sampling and execution of a single trajectory. |
|---|---|

Algorithm 1: Levin tree search.

```
1  def LevinTS()
2     V := ∅
3     F := {n₀}
4     while F ≠ ∅
5        n := argmin_{n∈F} d₀(n)/π(n)
6        F := F \ {n}
7        s := T(n)
8        if s ∈ Sᵍ
9           return true
10       if is_Markov(π)
11          if ∃n′ ∈ V : (T(n′) = s) ∧ (π(n′) ≥ π(n))
12             # s has already been visited with
13             # a higher probability: State cut
14             continue
15       V := V ∪ {n′}
16       F := F ∪ C(n)
17    return false
```

Algorithm 2: Sampling and execution of a single trajectory.

```
def sample_traj(depth)
   n := n₀
   for d := 0 to depth
      if T(n) ∈ Sᵍ
         return true
      a ∼ π(.|n)
      n := na
   return false
```

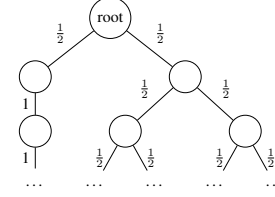

Figure 1: A 'chain-and-bin' tree.

## 3 Levin tree search: policy-guided enumeration

First, we show that merely expanding nodes by decreasing order of their probabilities can fail to reach a goal state of non-zero probability.

**Theorem 1.** The version of TS that chooses at iteration $k$ the node $n_k := \text{argmax}_{n\in\mathcal{F}_k} \pi(n)$ may never expand any node of the target set $\mathcal{N}^g$, even if $\forall n \in \mathcal{N}^g, \pi(n) > 0$.

*Proof.* Consider the tree in Fig. 1. Under the left child of the root is an infinite 'chain' in which each node has probability $1/2$. Under the right child of the root is an infinite binary tree in which each node has two children, each of conditional probability $1/2$, and thus each node has probability $2^{-d}$. Before testing a node of depth at least 2 in the right-hand-side binary tree (with probability at most $1/4$), the search expands infinitely many nodes of probability $1/2$. Defining the target set as any set of nodes with individual probability at most $1/4$ proves the claim. □

To solve this problem, we draw inspiration from Levin search [Levin, 1973, Trakhtenbrot, 1984], which (in a different domain) penalizes the probability with computation time. Here, we take computation time to mean the depth of a node. The new Levin tree search (LevinTS) algorithm is a version of TS in which nodes are expanded in order of increasing costs $d_0(n)/\pi(n)$ (see Algorithm 1).

LevinTS also performs *state cuts* (see Lines 10–15 of Algorithm 1). That is, LevinTS does not expand node $n$ representing state $s$ if (i) the policy $\pi$ is Markovian, (ii) it has already expanded another node $n'$ that also represents $s$, and (iii) $\pi(n') \geq \pi(n)$. By performing state cuts only if these three conditions are met, we can show that LevinTS expands states in best-first order.

**Theorem 2.** LevinTS expands states in best-first order and at their lowest cost first.

*Proof.* Let us first consider the case where the policy is non-Markovian. Then, LevinTS does not perform state cuts (see Line 10 of Algorithm 1). Let $n_1$ and $n_2$ be two arbitrary different nodes (sequences of actions), with $\text{cost}(n_1) < \text{cost}(n_2)$. Let $n_{12}$ be the closest common ancestor of $n_1$ and $n_2$; it must exist since at least the root is one of their common ancestors. Then all nodes on the path from $n_{12}$ to $n_1$ have cost less than $\text{cost}(n_1)$ and thus than $\text{cost}(n_2)$, due to the monotonicity of $d_0$ and $\pi$ and thus of cost, which implies by recursion from $n_{12}$ that all these nodes and thus also $n_1$ are expanded before $n_2$. Hence, if $T(n_1) = T(n_2)$, this proves that all states are visited first at their lowest cost. Furthermore, if $T(n_1) \neq T(n_2)$, this proves that states of lower cost are visited first.

Now, if the policy is Markovian, then we need to show that state cuts do not prevent best-first order and lowest cost. Let $n_1$ and $n_2$ be two nodes representing the same state $s$, where $n_1$ is expanded

before $n_2$. Assume that no cut has been performed before $n_2$ is expanded. First, since no cuts were performed, we have from the non-Markovian case that $\frac{d_0(n_1)}{\pi(n_1)} \leq \frac{d_0(n_2)}{\pi(n_2)}$. Secondly, consider a sequence of actions $a_{1:k}$ taken after state $s$, and let $n_{1k} = n_1 a_{1:k}$ be the node reached after taking $a_{1:k}$ starting from $n_1$ and similarly for $n_{2k}$. Since the environment is deterministic, this sequence leads to the same state $s_k$, whether starting from $n_1$ or from $n_2$. Since the policy is Markovian, $\pi(n_{1k}|n_1) = \pi(n_{2k}|n_2)$. Then from the condition (iii) of state cuts,

$$\text{if } \pi(n_1) \geq \pi(n_2), \quad \frac{d_0(n_{1k})}{\pi(n_{1k})} = \frac{d_0(n_1)}{\pi(n_1)} \frac{1}{\pi(n_{1k}|n_1)} + \frac{k}{\pi(n_1)\pi(n_{1k}|n_1)}$$
$$\leq \frac{d_0(n_2)}{\pi(n_2)} \frac{1}{\pi(n_{1k}|n_1)} + \frac{k}{\pi(n_2)\pi(n_{1k}|n_1)} = \frac{d_0(n_{2k})}{\pi(n_{2k})},$$

so the state $s_k$ has a lower or equal cost below $n_1$ than below $n_2$. Since this holds for any such $a_{1:k}$, $n_2$ can be safely cut, and by recurrence all cuts preserve the best-first ordering and lowest costs of states. The rest of the proof is as in the non-Markovian case. $\square$

LevinTS's cost function allows us to provide the following guarantee, which is an adaptation of Levin search's theorem [Solomonoff, 1984] to tree search problems.

**Theorem 3.** Let $\mathcal{N}^g$ be a set of target nodes, then LevinTS with a policy $\pi$ ensures that the number of node expansions $N(\text{LevinTS}, \mathcal{N}^g)$ before reaching any of the target nodes is bounded by

$$N(\text{LevinTS}, \mathcal{N}^g) \leq \min_{n \in \mathcal{N}^g} \frac{d_0(n)}{\pi(n)}\,.$$

*Proof.* From Theorem 2, the first state of $\mathcal{S}^g$ to be expanded is the one of lowest cost, and with one of the nodes of lowest cost, that is, with cost $c := \min_{n \in \mathcal{N}^g} d_0(n)/\pi(n)$. Let $\mathcal{T}_c$ be the current search tree when $n^g$ is being expanded. Then all nodes in $\mathcal{T}_c$ that have been expanded up to now have at most cost $c$. Therefore at all leaves $n \in \mathcal{L}(\mathcal{T}_c)$ of the current search tree, $d_0(n)/\pi(n) \leq c$. Since each node is expanded at most once (each sequence of actions is tried at most once) the number of nodes expanded by LevinTS until node $n^g$ is at most

$$N(\text{LevinTS}, \mathcal{N}^g) = |\mathcal{N}(\mathcal{T}_c)| \leq \sum_{n \in \mathcal{L}(\mathcal{T}_c)} d_0(n) \leq \sum_{n \in \mathcal{L}(\mathcal{T}_c)} \pi(n)c \leq c = \min_{n \in \mathcal{N}^g} \frac{d_0(n)}{\pi(n)}$$

where the first inequality is because each leaf of depth $d_0$ has at most $d_0$ ancestors, the second inequality follows from $d_0(n)/\pi(n) \leq c$, and the last inequality is because $\sum_{n \in \mathcal{L}(\mathcal{T}_c)} \pi(n) \leq 1$, which follows from $\sum_{n' \in \mathcal{C}(n)} \pi(n') = \pi(n)$, that is, each parent node splits its probability among its children, and the root has probability 1. $\square$

The upper bound of Theorem 3 is tight within a small factor for a tree like in Fig. 1, and is almost an equality when the tree splits at the root into multiple chains.

## 4 Luby tree search: policy-guided unbounded sampling

**Multi-sampling** When a good upper bound $d_{\max}$ is known on the depth of a subset of the target nodes with large cumulative probability, a simple idea is to sample trajectories according to $\pi$ (see Algorithm 2) of that maximum depth $d_{\max}$ until a solution is found, if one exists. Call this strategy multiTS (see Algorithm 3). We can then provide the following straightforward guarantee.

**Theorem 4.** The expected number of node expansions before reaching a node in $\mathcal{N}^g$ is bounded by

$$\mathbb{E}[N(\text{multiTS}(\infty, d_{\max}), \mathcal{N}^g)] \leq \frac{d_{\max}}{\pi_{d_{\max}}^+}\,, \qquad \pi_{d_{\max}}^+ := \sum_{\substack{n \in \mathcal{N}^g \\ d_0(n) \leq d_{\max}}} \pi(n)\,.$$

*Proof.* Remembering that a tree search algorithm does not expand children of target nodes, the result follows from observing that $\mathbb{E}[N(\text{multiTS}, \mathcal{N}^g)]$ is the expectation of a geometric distribution with success probability $\pi_{d_{\max}}^+$ where each failed trial takes exactly $d_{\max}$ node expansions and the success trial takes at most $d_{\max}$ node expansions. $\square$

Algorithm (3) Sampling of `nsims` trajectories of fixed depths $d_{\max} \in \mathbb{N}_1$.

```
def multiTS(nsims, dmax)
  for k := 1 to nsims
    if sample_traj(dmax)
      return true
  return false
```

Algorithm (4) Sampling of `nsims` trajectories of depths that follow A6519, with optional coefficient $d_{\min} \in \mathbb{N}_1$.

```
def LubyTS(nsims, dmin=1)
  for k := 1 to nsims
    if sample_traj(dmin * A6519(k))
      return true
  return false
```

This strategy can have an important advantage over LevinTS if there are many target nodes within depth bounded by $d_{\max}$ with small individual probability but large cumulative probability.

The drawback is that if no target node has a depth shorter than the bound $d_{\max}$, this strategy will never find a solution (the expectation is infinite), even if the target nodes have high probability according to the policy $\pi$. Ensuring such target nodes can be always found leads to the LubyTS algorithm.

**LubyTS** Suppose we are given a randomized program $\rho$, that has an unknown distribution $p$ over the halting times (where halting means solving an underlying problem). We want to define a strategy that can restart the program multiple times and run it each time with a different allowed running time so that it halts in as little cumulative time as possible in expectation. Luby et al. [1993] prove that the optimal strategy is to run $\rho$ for running times of fixed lengths $t_p$ optimized for $p$; then either the program halts within $t_p$ steps, or it is forced to stop and is restarted for another $t_p$ steps and so on. This strategy has an expected running time of $\ell_p$, with $\frac{L_p}{4} \leq \ell_p \leq L_p = \min_{t \in \mathbb{N}_1} \frac{t}{q(t)}$ where $q$ is the cumulative distribution function of $p$. Luby et al. [1993] also devise a *universal* restarting strategy based on a special sequence[2] of running times:

$$1\ 1\ 2\ 1\ 1\ 2\ 4\ 1\ 1\ 2\ 1\ 1\ 2\ 4\ 8\ 1\ 1\ 2\ 1\ 1\ 2\ 4\ 1\ 1\ 2\ 1\ 1\ 2\ 4\ 8\ 16\ 1\ 1\ 2\ldots$$

They prove that the expected running time of this strategy is bounded by $192\ell_p(\log_2 \ell_p + 5)$ and also prove a lower bound of $\frac{1}{8}\ell_p \log_2 \ell_p$ for any universal restarting strategy. We propose to use instead the sequence[3] A6519:

$$1\ 2\ 1\ 4\ 1\ 2\ 1\ 8\ 1\ 2\ 1\ 4\ 1\ 2\ 1\ 16\ 1\ 2\ 1\ 4\ 1\ 2\ 1\ 8\ 1\ 2\ 1\ 4\ 1\ 2\ 1\ 32\ 1\ 2\ldots$$

which is simpler to compute and for which we can prove the following tighter upper bound.

**Theorem 5.** For all distributions $p$ over halting times, the expected running time of the restarting strategy based on A6519 is bounded by $\min_t t + \frac{t}{q(t)}\left(\log_2 \frac{t}{q(t)} + 6.1\right)$, where $q$ is the cumulative distribution of $p$.

The proof is provided in Appendix B. We can easily import the strategy described above into the tree search setting (see Algorithm 4), and provide the following result.

**Theorem 6.** Let $\mathcal{N}^g$ be the set of target nodes, then LubyTS$(\infty, 1)$ with a policy $\pi$ ensures that the expected number of node expansions before reaching a target node is bounded by

$$\mathbb{E}[N(\text{LubyTS}(\infty, 1), \mathcal{N}^g)] \leq \min_{d \in \mathbb{N}_1} d + \frac{d}{\pi_d^+}\left(\log_2 \frac{d}{\pi_d^+} + 6.1\right), \qquad \pi_d^+ := \sum_{\substack{n \in \mathcal{N}^g \\ d_0(n) \leq d}} \pi(n),$$

where $\pi_d^+$ is the cumulative probability of the target nodes with depth at most $d$.

*Proof.* This is a straightforward application of Theorem 5: The randomized program samples a sequence of actions from the policy $\pi$, the running time $t$ becomes the depth $d_0(n)$ of a node $n$, the probability distribution $p$ over halting times becomes the probability of reaching a target node of depth $t$, $p(t) = \sum_{\{n \in \mathcal{N}^g, d_0(n) = t\}} \pi(n)$, and the cumulative distribution function $q$ becomes $\pi_d^+$. $\square$

Compared to Theorem 4, the cost of adapting to an unknown depth is an additional factor $\log(d/\pi_d^+)$. The proof of Theorem 5 suggests that the term $\log d$ is due to not knowing the lower bound on $d$, and the term $-\log \pi_d^+$ is due to not knowing the upper bound. If a good lower bound $d_{\min}$ on the average solution length is known, one can also multiply A6519$(n)$ by $d_{\min}$ to avoid sampling too short trajectories as in Algorithm 4; this may lessen the factor $\log d$ while still guaranteeing that a solution can be found if one of positive probability exists. In particular, in the tree search domain, the sequence A6519 samples trajectories of depth 1 half of the time, which is wasteful. Conversely, in general it is not possible to cap $d$ at some upper bound, as this may prevent finding a solution as for multiTS. Hence the factor $-\log \pi_d^+$ remains, which is unfortunate since $\pi_d^+$ can easily be exponentially small with $d$.

## 5 Strengths and weaknesses of LevinTS and LubyTS

Consider a "needle-in-the-haystack problem" represented by a perfect full and infinite binary search tree where all nodes $n$ have probability $\pi(n) = 2^{-d(n)}$. Suppose that the set $\mathcal{N}^g$ of target nodes contains a single node $n^g$ at some depth $d$. According to Theorem 3, LevinTS needs to expand no more than $d_0(n^g)2^{d(n^g)}$ nodes before expanding $n^g$. For this particular tree, the number of expansions is closer to $2^{d(n^g)+1}$ since there are only at most $2^{d(n^g)-1}$ nodes with cost lower or equal to $\mathrm{cost}(n^g)$. Theorem 6 and the matching-order lower bound of [Luby et al., 1993] suggest LubyTS may expand in expectation $O(d(n^g)^2 2^{d(n^g)})$ nodes to reach $n^g$. This additional factor of $d(n)^2$ compared to LevinTS is a non-negligible price for needle-in-a-haystack searches. For multiTS, if the depth bound $d_{\max}$ is larger than $d_0(n^g)$, then the expected search time is at most and close to $d_{\max}2^{d(n^g)}$, which is a factor $d(n)$ faster than LubyTS, unless $d_{\max} \gg d(n^g)$.

Now suppose that the set of target nodes is composed of $2^{d-1}$ nodes, all at depth $d$. Since all nodes at a given depth have the same probability, LevinTS will expand at least $2^d$ and at most $2^{d+1}$ nodes before expanding any of the target nodes. By contrast, because the cumulative probability of the target nodes at depth $d$ is $1/2$, LubyTS finds a solution in $O(d \log d)$ node expansions, which is an exponential gain over LevinTS. For multiTS it would be $d_{\max}$, which can be worse than $d \log d$ due to the need for a large enough $d_{\max}$.

LevinTS can perform state cuts if the policy is Markovian, which can substantially reduce the algorithm's search effort. For example, suppose that in the binary tree above every left child represents the same state as the root and thus is cut off from the search tree, leaving in effect only $2d$ nodes for any depth $d$. If the target set contains only one node at some depth $d$, even when following a uniform policy, LevinTS expands only those $2d$ nodes. By contrast, LubyTS expands in expectation more than $O(2^d)$ nodes. LevinTS has a memory requirement that grows linearly with the number of nodes expanded, as well as a log factor in the computation time due to the need to maintain a priority queue to sort the nodes by cost. By contrast, LubyTS and multiTS have a memory requirement that grows linearly with the solution depth, as they only need to store in memory the trajectory sampled. LevinTS's memory cost could be alleviated with an iterative deepening [Korf, 1985] variant with transposition table [Reinefeld and Marsland, 1994].

## 6 Mixing policies and avoiding zero probabilities

For both LevinTS and LubyTS, if the provided policy $\pi$ incorrectly assigns a probability too close to 0 to some sequences of actions, then the algorithm may never find the solution. To mitigate such outcomes, it is possible to 'mix' the policy with the uniform policy so that the former behaves slightly more like the latter. There are several ways to achieve this, each with their own pros and cons.

**Bayes mixing of policies** If $\pi_1$ and $\pi_2$ are two policies, we can build their Bayes average $\pi_{12}$ with prior $\alpha \in [0, 1]$ and $1 - \alpha$ such that for all sequence of actions $a_{1:t}$, $\pi_{12}(a_{1:t}) = \alpha\pi_1(a_{1:t}) + (1 - \alpha)\pi_2(a_{1:t})$. The conditional probability of the next action is given by

$$\pi_{12}(a_t|a_{<t}) = w_1(a_{<t})\pi_1(a_t|a_{<t}) + w_2(a_{<t})\pi_2(a_t|a_{<t})$$

$$\text{with } w_1(a_{<t}) = 1 - w_2(a_{<t}) = \frac{\alpha\pi_1(a_{<t})}{\alpha\pi_1(a_{<t}) + (1 - \alpha)\pi_2(a_{<t})} = \alpha\frac{\pi_1(a_{<t})}{\pi_{12}(a_{<t})},$$

where $w_1(a_{<t})$ is the 'posterior weight' of the policy $\pi_1$ in $\pi_{12}$. This ensures that for all nodes $n$, $\pi_{12}(n) \geq \alpha\pi_1(n)$ and $\pi_{12}(n) \geq (1-\alpha)\pi_2(n)$ which leads to the following refinement for Theorem 3 for example (and similarly for LubyTS):

$$N(\text{LevinTS}, \mathcal{N}^g) \leq \min\left\{\frac{1}{\alpha}\min_{n\in\mathcal{N}^g}\frac{d_0(n)}{\pi_1(n)}, \frac{1}{1-\alpha}\min_{n\in\mathcal{N}^g}\frac{d_0(n)}{\pi_2(n)}\right\}.$$

In particular, with $\alpha = 1/2$, LevinTS with $\pi_{12}$ is within a factor 2 of the best between LevinTS with $\pi_1$ and LevinTS with $\pi_2$. More than two policies can be mixed together, leading for example to a factor $K$ compared to the best of $K$ policies when all prior weights are equal. This is very much like running several instances of LevinTS in parallel, each with its own policy, except that (weighted) time sharing is done automatically. For example, if the provided policy $\pi$ is likely to occasionally assign too low probabilities, one can run LevinTS with a Bayes mixture of $\pi$ and the uniform policy, with a prior weight $\alpha$ closer to 1 if $\pi$ is likely to be far better than the uniform policy for most instances.

**Local mixing of policies, fixed rate** Bayes mixing of two policies splits the search into 2 (mostly) independent searches. But one may want to mix at a more 'local' level: Along a trajectory $a_{1:t}$, if the provided policy $\pi$ assigns high probability to almost all actions but a very low probability to a few ones, we may want to use a different policy just for these actions, and not for the whole trajectory. Thus, given two policies $\pi_1$ and $\pi_2$ and $\varepsilon \in [0, 1]$, the local-mixing policy $\pi_{12}$ is defined through its conditional probability $\pi_{12}(a_t|a_{<t}) := \varepsilon\pi_1(a_t|a_{<t}) + (1-\varepsilon)\pi_2(a_t|a_{<t})$. Then for all $a_{1:t}$,

$$\pi_{12}(a_{1:t}) \geq \underbrace{\varepsilon^{|\mathcal{K}_1|}(1-\varepsilon)^{t-|\mathcal{K}_1|}}_{\text{penalty}}\prod_{k\in\mathcal{K}_1}\pi_1(a_k|a_{<k})\prod_{k\notin\mathcal{K}_1}\pi_2(a_k|a_{<k}),$$

where $\mathcal{K}_1$ is the set of steps $k$ where $\pi_1(a_k|a_{<k}) > \pi_2(a_t|a_{<k})$. This can be interpreted as 'At each step $t$, $\pi$ must pay a factor of $\varepsilon$ to use policy $\pi_1$ or a factor of $1-\varepsilon$ to use $\pi_2$'. This works well for example if $\varepsilon \approx 0$ and $\mathcal{K}_1$ is small, that is, the policy $\pi_2$ is used most of the time. For example, $\pi_1$ can be the uniform policy, $\pi_1(a_t|a_{<t}) = 1/|\mathcal{A}|$, and $\pi_2$ is a given policy that may sometimes be wrong.

**Local mixing, varying rate** The problem with the previous approach is that $\varepsilon$ needs to be fixed in advance. For a depth $d$, a penalty of the number of node expansions of $1/(1-\varepsilon)^d \approx e^{\varepsilon d}$ is large as soon as $d > 1/\varepsilon$. If no good bound on $d$ is known, one can use a more adaptive $1-\varepsilon_d(a_{1:t}) = (t/(t+1))^\gamma$ with $\gamma \geq 0$: This gives $\prod_{k=1}^{t}(t/(t+1))^\gamma = 1/(t+1)^\gamma$, which means that the maximum price to pay to use only the policy $\pi_2$ for all the $t$ steps is at most $(t+1)^\gamma$, and the price to pay each step the policy $\pi_1$ is used is approximately $(t+1)/\gamma$. The optimal value of $\varepsilon$ can also be learned automatically using an algorithm such as Soft-Bayes [Orseau et al., 2017] where the 'experts' are the provided policies, but this may have a large probability overhead for this setup.

## 7 Experiments: computer-generated Sokoban

We test our algorithms on 1,000 computer-generated levels of Sokoban [Racanière et al., 2017] of 10x10 grid cells and 4 boxes.[4] For the policy, we use a neural network pre-trained with A3C (details on the architecture and the learning procedure are in Appendix A). We picked the best performing network out of 4 runs with different learning rates. Once the network is trained, we compare the different algorithms using the same network's fixed Markovian policy. Note that for each new level, the goal states (and thus target set) are different, whereas the policy does not change (but still depends on the state). We test the following algorithms and parameters: LubyTS(256,1), LubyTS(256,32), LubyTS(512, 32), multiTS(1, 200), multiTS(100, 200), multiTS(200, 200), LevinTS. Excluding the small values (i.e., `nsims = 1` and $d_{\min} = 1$), the parameters were chosen to obtain a total number of expansions within the same order of magnitude. In addition to the policy trained with A3C, we tested LevinTS, LubyTS, and multiTS with a variant of the policy in which we add 1% of noise to the probabilities output of the neural network. That is, these variants use the policy $\tilde{\pi}(a|n) = (1-\varepsilon)\pi(a|n) + \varepsilon\frac{1}{4}$ where $\pi$ is the network's policy and $\varepsilon = 0.01$, to guide their search. These variants are marked with the symbol (*) in the table of results. We compare our policy tree search methods with a version of the LAMA planner [Richter and Westphal, 2010] that uses the lazy version of GBFS with preferred operators and queue alternation with the FF heuristic. This version of

Table 1: Comparison of different solvers on the 1000 computer-generated levels of Sokoban. For randomized solvers (shown at the top part of the table), the results are aggregated over 5 random seeds ($\pm$ indicates standard deviation). (*) Uses $\tilde{\pi}$ with $\varepsilon = 0.01$.

| Algorithm | Solved | Avg. length | Max. length | Total expansions |
|---|---|---|---|---|
| Uniform | 88 | 19 | 59 | 94,423,278 |
| LubyTS(256, 1) | $753 \pm 5$ | $41.0 \pm 0.6$ | $228 \pm 18.6$ | $63,8481 \pm 2,434$ |
| LubyTS(256, 32) | $870 \pm 2$ | $48.4 \pm 0.9$ | $1,638.4 \pm 540.7$ | $6,246,293 \pm 73,382$ |
| LubyTS(512, 32) | $884 \pm 4$ | $54.8 \pm 4.2$ | $3,266.6 \pm 1,287.8$ | $11,515,937 \pm 211,524$ |
| LubyTS(512, 32) (*) | $896 \pm 2$ | $50.7 \pm 2.5$ | $1,975.6 \pm 904.5$ | $10,730,753 \pm 164,410$ |
| MultiTS(1, 200) | $669 \pm 5$ | $41.3 \pm 0.6$ | $196.4 \pm 2.2$ | $93,768 \pm 925$ |
| MultiTS(100, 200) | $866 \pm 4$ | $47.8 \pm 0.5$ | $199.4 \pm 0.5$ | $3260536 \pm 57185$ |
| MultiTS(200, 200) | $881 \pm 1$ | $47.9 \pm 0.7$ | $196.4 \pm 2.3$ | $5,768,680 \pm 116,152$ |
| MultiTS(200, 200) (*) | $895 \pm 3$ | $48.8 \pm 0.4$ | $198.8 \pm 1$ | $5,389,534 \pm 45,085$ |
| LevinTS | 1,000 | 39.8 | 106 | 6,602,666 |
| LevinTS (*) | 1,000 | 39.5 | 106 | 5,026,200 |
| LAMA | 1,000 | 51.6 | 185 | 3,151,325 |

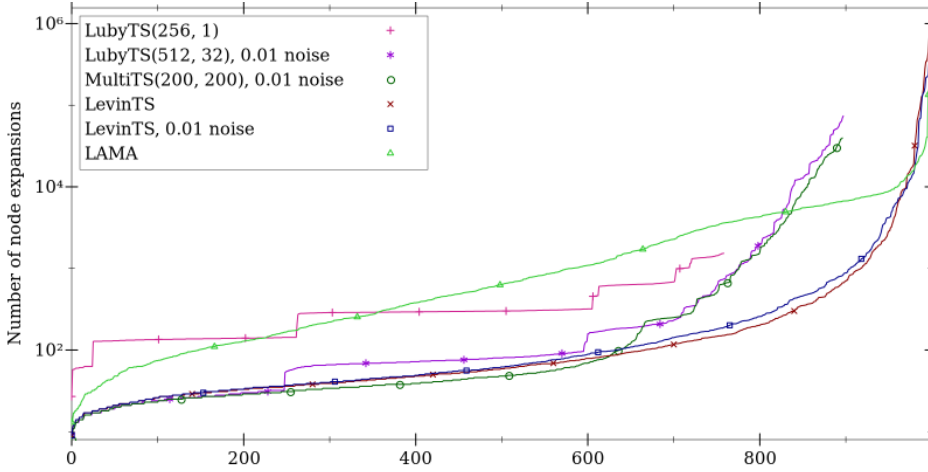

Figure 3: Node expansions for Sokoban on log-scale. The levels indices (x-axis) are sorted independently for each solver from the easiest to the hardest level. For clarity a typical run has been chosen for randomized solvers; see Table 1 for standard deviations.

LAMA is implemented in Fast Downward [Helmert, 2006], a domain-independent solver. We used this version of LAMA because it was shown to perform better than other state-of-the-art planners on Sokoban problems [Xie et al., 2012]. Moreover, similarly to our methods, LAMA searches for a solution of small depth rather than a solution of minimal depth.

Table 1 presents the number of levels solved ("Solved"), average solution length ("Avg. length"), longest solution length ("Max. length"), and total number of nodes expanded ("Total expansions"). The top part of the table shows the sampling-based randomized algorithms. In addition to the average values, we present the standard deviation of five independent runs of these algorithms. Since LevinTS and LAMA are deterministic, we present a single run of these approaches. Fig. 3 shows the number of nodes expanded per level by each method when the levels are independently sorted for each approach from the easiest to the hardest Sokoban level in terms of node expansions. The Uniform searcher (LevinTS with a uniform policy) with maximum 100,000 node expansions per level—and still with state cuts—can solve no more than 9% of the levels, which shows that the problem is not trivial.

For most of the levels, LevinTS (with the A3C policy) expands many fewer nodes than LAMA, but has to expand many more nodes on the last few levels. On 998 instances, the cumulative number of expansions taken by LevinTS is ~2.7e6 nodes while LAMA expands ~3.1e6 nodes. These numbers contrast with the number of expansions required by LevinTS (6.6e6) and LAMA (3.15e6) to solve all

1,000 levels. The addition of noise to the policy reduces the number of nodes expanded by LevinTS while solving harder instances at the cost of increasing the number of nodes expanded for easier problems (see the lines of the two versions of LevinTS crossing at the right-hand side of Fig. 3). Overall, noise reduces from 6.6e6 to 5e6 the total number of nodes LevinTS expands (see Table 1). LevinTS has to expand a large number of nodes for a small number of levels likely due to the training procedure used to derive the policy. That is, the policy is learned only from the 65% easiest levels solved after sampling single trajectories—harder levels are never solved during training. Nevertheless, LevinTS can still solve harder instances by compensating the lack of policy guidance with search.

The sampling-based methods have a hard time reaching 90% success, but still improves by more than 20% over sampling a single trajectory. LubyTS(256, 32) improves substantially over LubyTS(256, 1) since many solutions have length around 30 steps. LubyTS(256, 32) is as good as multiTS(200, 100) that uses a hand-tuned upper bound on the length of the solutions.

The solutions found by LevinTS are noticeably shorter (in terms of number of moves) than those found by LAMA. It is remarkable that LevinTS can find shorter solutions and expand fewer nodes than LAMA for most of the levels. This is likely due to the combination of good search guidance through the policy for most of the problems and LevinTS's systematic search procedure. By contrast, due to its sampling-based approach, LubyTS tends to find very long solutions.

Racanière et al. [2017] report different neural-network based solvers applied to a long sequence of Sokoban levels generated by the same system used in our experiments (although we use a different random seed to generate the levels, we believe they are of the same complexity). Racanière et al.'s primary goal was not to produce an efficient solver per se, but to demonstrate how an integrated neural-based learning and planning system can be robust to model errors and more efficient than an MCTS baseline. Their MCTS approach solves 87% of the levels within approximately 30e6 node expansions (25,000 per level for 870 levels, and 500 simulations of 120 steps for the remaining 130 levels). Although LevinTS had much stronger results in our experiments, we note that Racanière et al.'s implementation of MCTS commits to an action every 500 node expansions. By contrast, in our experimental setup, we assume that LevinTS solves the problem before committing to an action. This difference makes the results not directly comparable. Racanière et al.'s second solver (I2A) is a hybrid model-free and model-based planning using a LSTM-based recurrent neural network with more learning components than our approaches. I2A reaches 95% success within an estimated total of 5.3e6 node expansions (4,000 on average over 950 levels, and 30,000 steps for the remaining 50 unsolved levels; this counts the internal planning steps). For comparison, LevinTS with 1% noise solves all the levels within the same total time (999 for LevinTS without noise). Moreover, LevinTS solves 95% of the levels within a total of less than 160,000 steps, which is approximately 168 node expansions on average for solved levels, compared to the reported 4,000 for I2A. Moreover, it is also not clear how long it would take I2A to solve the remaining 5%.

## 8 Conclusions and future works

We introduced two novel tree search algorithms for single-agent problems that are guided by a policy: LevinTS and LubyTS. Both algorithms have guarantees on the number of nodes that they expand before reaching a solution (strictly for LevinTS, in expectation for LubyTS). LevinTS and LubyTS depart from the traditional heuristic approach to tree search by employing a policy instead of a heuristic function to guide search while still offering important guarantees.

The results on the computer-generated Sokoban problems using a pre-trained neural network show that these algorithms can largely improve through tree search upon the score of the network during training. Our results also showed that LevinTS is able to solve most of the levels used in our experiment while expanding many fewer nodes than a state-of-the-art heuristic search planner. In addition, LevinTS was able to find considerably shorter solutions than the planner.

The policy can be learned by various means or it can even be handcrafted. In this paper we used reinforcement learning to learn the policy. However, the bounds offered by the algorithms could also serve directly as metrics to be optimized while learning a policy; this is a research direction we are interested in investigating in future works.

**Acknowledgements** The authors wish to thank Peter Sunehag, Andras Gyorgy, Rémi Munos, Joel Veness, Arthur Guez, Marc Lanctot, André Grahl Pereira, and Michael Bowling for helpful discussions pertaining this research. Financial support for this research was in part provided by the Natural Sciences and Engineering Research Council of Canada (NSERC).

## Footnotes

[2] https://oeis.org/A182105.

[3] https://oeis.org/A006519. Gary Detlefs (ibid) notes that it can be computed with A6519($n$) := $((n \text{ XOR } n - 1) + 1)/2$ or with A6519($n$) := $(n \text{ AND } - n)$ where $-n$ is $n$'s complement to 2.

[4]The levels are available at `https://github.com/deepmind/boxoban-levels/unfiltered/test`.

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
