[Supplementary Material · supplementary-nips2018-tree-search.pdf]

# Single-Agent Policy Tree Search With Guarantees: Supplementary Material

**Laurent Orseau**
DeepMind, London, UK
lorseau@google.com

**Levi H. S. Lelis**[*]
Universidade Federal de Viçosa, Brazil
levi.lelis@ufv.br

**Tor Lattimore**
DeepMind, London, UK
lattimore@google.com

**Théophane Weber**
DeepMind, London, UK
theophane@google.com

---

[*]This work was carried out while L. H. S. Lelis was at the University of Alberta, Canada.

Figure 4: Learning curves of A3C for the 4 chosen learning rates (4e-4, 2e-4, 1e-4, 5e-5) on the Sokoban level generator.

## A Network architecture and learning protocol

The network takes as input a 10x10x4 grid where the last dimension is for a binary encoding of the different attributes (wall, man, goal, box), which is passed through 2 convolutional layers ($4 \times 4$ with 64 channels, followed by $3 \times 3$ with 64 channels as well), followed by a fully connected layer of 512 ReLU units. The output layer provides logits for the 4 actions (up, down, left, right). Training is performed using A3C [Mnih et al., 2016] with a reward function giving a reward of -0.1 per step, +1 per box on a goal and -1 for the converse action, and +10 for solving the level (all boxes on goals), with a discount factor of 0.99; the optimizer used is RMSProp [Tieleman and Hinton, 2012] (no momentum, epsilon 0.1, decay 0.99), with entropy regularization of 0.005. During training, at each episode, the learner performs a single trajectory of length 100 (like multiTS(1, 100)), receives the corresponding rewards, then moves on to the next episode. A single level is (very likely) never seen twice during training. Similarly, it is very unlikely that a level of the 1000 test levels was seen during training. We take the best performing network, which solves around 65% of the levels when sampling a single sequence of actions. The network is trained for 3.5e9 steps (node expansions), which can seem to be a lot, however notice that this is equivalent to fully searching a *single* level of Sokoban (without state cuts) uniformly with 4 actions up to depth 16 (given that solutions are usually of depth more than 30). The learning process was repeated for 4 learning rates (4e-4, 2e-4, 1e-4, 5e-5) (see Fig. 4).

## B Another universal restarting strategy for Las Vegas programs

We use the sequence[5] of runtimes $f(n) := \text{A6519}(n)$:

$$1\ 2\ 1\ 4\ 1\ 2\ 1\ 8\ 1\ 2\ 1\ 4\ 1\ 2\ 1\ 16\ 1\ 2\ 1\ldots$$

$$\text{For all } n \in \mathbb{N}_1 : f(n) := \begin{cases} 1 & \text{if } n \text{ is odd,} \\ 2f(n/2) & \text{o.w.} \end{cases}$$

It has the 'fractal' property that $f(k2^n) = 2^n f(k)$ (since $f(k2^n) = 2f(k2^{n-1}) = \ldots = 2^n f(k2^0)$), for $k \in \mathbb{N}_1$ and $n \in \mathbb{N}_0$, and it follows that $f(2^n) = 2^n$ and $f(k2^n) \geq 2^n$.

At iteration $n$, the Las Vegas program is run for $f(n)$ steps. For all $t > 0$, if $f(n) \geq t$, then it has a probability at least $q(t)$ of halting, otherwise it does not halt and is forcibly stopped after $f(n)$ computations steps. Let $\hat{t} := 2^{\lceil \log_2 t \rceil}$ be the smallest power of 2 greater than or equal to $t$. Then Lemma 8 below tells us that for $c < \hat{t}$ we have that $f(k\hat{t} + c) = f(c) \leq \hat{t}/2 < t$, that is, between two consecutive factors of $\hat{t}$, $f(n) < t$.

Let $p_{\text{halt}}(n)$ denote the probability that the algorithm halts exactly at the $n$th run, and take $1 \leq c < \hat{t}$ and $k \geq 0$, then the expected number of computation steps $L$ (sum of the lengths of the runs) before

halting is given by:

$$L_{\text{univ}}(p) := \sum_{n=1}^{\infty} \left[ t\, p_{\text{halt}}(n) + (1 - p_{\text{halt}}(n))f(n) \right] \underbrace{\prod_{j=1}^{n-1} (1 - p_{\text{halt}}(j))}_{\substack{\text{probability of} \\ \text{not halting before run } n}} .$$

where $p_{\text{halt}}(n) = 0$ when $f(n) < t$, and $p_{\text{halt}}(n) = q(t)$ otherwise.

We restate Theorem 5 more precisely:

**Theorem 7.** For all distributions $p$ over halting times, the expected runtime of the universal restarting strategy based on $A6519$ is bounded by

$$L_{\text{univ}}(p) \leq \min_t t + \frac{t}{q(t)} \left( \log_2 \frac{t}{q(t)} + 6.1 \right) ,$$

where $q$ is the cumulative distribution of $p$.

*Proof of Theorems 5 and 7.* At step $n$, if $k$ is the number of past runs where $f(m) \geq \hat{t}$ (with $m < n$), then $\prod_{j=1}^{n-1}(1 - p_{\text{halt}}(j)) = (1 - q(t))^k$ then with $1 \leq c < \hat{t}$ and $\gamma := 1 - q(t)$:

$$L_{\text{univ}}(p) = \sum_{n=0}^{\infty} \begin{cases} \gamma^k f(n) & \text{if } n = k\hat{t} + c \quad (\text{i.e., } f(n) < t) \\ \gamma^k pt + \gamma^{k+1} f(n) & \text{if } n = k\hat{t} + \hat{t}, \end{cases}$$

$$= \sum_{n=0}^{\infty} \begin{cases} \gamma^k f(c) & \text{if } n = k\hat{t} + c \\ \gamma^k pt + \gamma^{k+1}\hat{t} f(k+1) & \text{if } n = k\hat{t} + \hat{t}. \end{cases}$$

where we used $f((k+1)\hat{t}) = \hat{t} f(k+1)$ (remembering that $\hat{t}$ is a power of 2) and Lemma 8 for $f(k\hat{t} + c) = f(c)$. Since $f(n) = f(c) < t$ when $n = k\hat{t} + c$, we can decompose $L_{\text{univ}}(p)$ into the steps where $f(n) < t$ and the rest:

$$L_{\text{univ}}(p) = L^< + L^\geq$$

$$L^< := \sum_{k=0}^{\infty} \gamma^k \sum_{c=1}^{\hat{t}-1} f(c) = \frac{1}{1 - \gamma} \sum_{c=1}^{\hat{t}-1} f(c) = \frac{\hat{t}}{2q(t)} \log_2 \hat{t} \quad (\text{Lemma 9})$$

$$L^\geq := \sum_{k=0}^{\infty} \gamma^k (1 - \gamma)t + \gamma^{k+1}\hat{t} f(k+1) = t + \hat{t} \sum_{k=1}^{\infty} \gamma^k f(k)$$

$$\leq t + \frac{\hat{t}}{q(\hat{t})} \left( \frac{1}{e} + \frac{1}{\ln 2} + \frac{1}{2} \log_2 \ln 16 + \frac{1}{2} \log_2 \frac{1}{q(\hat{t})} \right)$$

where we used Lemma 13 on the last line with $\gamma = 1 - q(t)$. Finally, since $\hat{t} = 2^{\lceil \log_2 t \rceil} \leq 2t$ and $q(\hat{t}) \geq q(t)$ and $\lceil \log_2 t \rceil \leq \log_2 t + 1$:

$$L \leq t + \frac{t}{q(t)} \left( \log_2 t + 1 + \frac{2}{e} + \frac{2}{\ln 2} + \log_2 \ln 16 + \log_2 \frac{1}{q(t)} \right)$$

$$\leq t + \frac{t}{q(t)} \left( \log_2 \frac{t}{q(t)} + 6.1 \right)$$

which proves the result. $\square$

**Lemma 8.** For $f = A6519$, with $k \in \mathbb{N}_0, n \in \mathbb{N}_0, a \in \mathbb{N}_1, b \in \mathbb{N}_0$ and $a2^b < 2^n$, and with $a$ odd, then

$$f(k2^n + a2^b) = f(a2^b) = 2^b .$$

*Proof.* Since $a$ is odd, then so is $k2^{n-b}+a$, and so $f(k2^n+a2^b) = f(2^b(k2^{n-b}+a)) = 2^b f(k2^{n-b}+a) = 2^b$. $\square$

Hence, for all numbers between two adjacent factors of $2^n$, $f(k2^n + c) = f(c) \leq 2^{n-1}$.

**Lemma 9.** For $n \in \mathbb{N}_1$ and $f =$A6519,

$$\sum_{c=1}^{2^n-1} f(c) = n2^{n-1}.$$

*Proof.* If $n \geq 1$ and using Lemma 8 again at $2^{n-1}$:

$$\sum_{c=1}^{2^n-1} f(c) = \sum_{c=1}^{2^{n-1}-1} f(c) + f(2^{n-1}) + \sum_{c=2^{n-1}+1}^{2^n-1} f(c)$$

$$= 2^{n-1} + 2 \sum_{c=1}^{2^{n-1}-1} f(c)$$

$$= \ldots = 2^0 2^{n-1} + 2^1 2^{n-2} + 2^2 2^{n-3} + \ldots + 2^{n-1} 2^0 + 2^n \sum_{c=1}^{2^0-1} f(c)$$

$$= n2^{n-1}.$$

$\square$

**Lemma 10.** Let $f =$A6519, then for $k \in \mathbb{N}_1, n \in \mathbb{N}_0, c \in \mathbb{N}_0$:

$$f(k) = 2^n \quad \Leftrightarrow \quad k = (2c+1)2^n.$$

*Proof.* Since any number $k$ can be uniquely written in the form $k = (2c+1)2^a$, and $f((2c+1)2^a) = 2^a f(2c+1) = 2^a$ with $a \in \mathbb{N}_0$, then $f(k) = 2^n \Leftrightarrow a = n$. $\square$

**Lemma 11.** For $\gamma \in [0, 1)$,

$$\sum_{n=0}^{\infty} 2^n \gamma^{2^n} \leq \frac{1}{\ln \frac{1}{\gamma}} \left( \frac{1}{e} + \frac{\gamma}{\ln 2} \right).$$

*Proof.* Let $h(x) := 2^x \gamma^{2^x}$ for $x \in \mathbb{R}$, then $h'(x) = \ln(2)2^x \gamma^{2^x} (2^x \ln \gamma + 1)$ where $h'(x_0) = 0$ for the unique $x_0$ such that $2^{x_0} = \frac{1}{\ln \frac{1}{\gamma}}$ and since $\ln \gamma < 0$, we have that $h'(x)$ is positive for $x < x_0$ and negative for $x > x_0$. Thus $h$ is unimodal, and since furthermore $h(x)$ is positive the sum can be upper bounded by the integral of the continuous function plus its maximum:

$$\sum_{n=0}^{\infty} h(n) \leq \int_0^{\infty} h(x)\mathrm{d}x + \max_x h(x),$$

$$\max_x h(x) = h(x_0) = \frac{1}{\ln \frac{1}{\gamma}} \frac{1}{e},$$

$$\int_0^{\infty} 2^x \gamma^{2^x} \mathrm{d}x = \frac{1}{\ln 2} \int_0^{\infty} 2^x \ln 2 \gamma^{2^x} \mathrm{d}x = \frac{1}{\ln 2} \int_1^{\infty} \gamma^y \mathrm{d}y = \frac{\gamma}{\ln 2 \ln \frac{1}{\gamma}},$$

where we used integration by substitution. Adding the two terms finishes the proof. $\square$

**Lemma 12.** For $\gamma \in [0, 1)$ and $a \geq 1$:

$$\sum_{n=0}^{\infty} \gamma^{2^n} \quad \leq \gamma \left\lceil \log_2 \frac{1}{\log_2 \frac{1}{\gamma}} \right\rceil + 1 \quad \leq \log_2 \frac{1}{\ln \frac{1}{\gamma}} + \log_2 \ln 16.$$

*Proof.* Let $N = \min\left\{n \in \mathbb{N}_0 : \gamma^{2^N} \leq \frac{1}{2}\right\} = \left\lceil \log_2 \frac{1}{\log_2 \frac{1}{\gamma}} \right\rceil$, then

$$\sum_{n=0}^{\infty} \gamma^{2^n} = \sum_{n=0}^{N-1} \gamma^{2^n} + \sum_{n=N}^{\infty} \gamma^{2^n}$$

$$\leq N\gamma + \sum_{n=0}^{\infty} \left(\gamma^{2^N}\right)^{2^n} \leq N\gamma + \sum_{n=0}^{\infty} 2^{-2^n} \leq N\gamma + 1$$

$$\leq \left\lceil \log_2 \frac{1}{\log_2 \frac{1}{\gamma}} \right\rceil + 1$$

$$\leq \log_2 \frac{1}{\log_2 \frac{1}{\gamma}} + 2 \,.$$

Extracting $\log_2 \ln 2$ finishes the proof. $\qquad\square$

**Lemma 13.** Let $f = $A6519 and $\gamma \in [0, 1)$. Then

$$\sum_{k=1}^{\infty} \gamma^k f(k) \leq \frac{1}{1-\gamma}\left(\frac{1}{e} + \frac{1}{\ln 2} + \frac{1}{2}\log_2 \ln 16 + \frac{1}{2}\log_2 \frac{1}{1-\gamma}\right) \,.$$

*Proof.* Since $f(n)$ is a power of 2 for all $n \in \mathbb{N}_1$, we regroup the runs by powers of 2:

$$\sum_{k=1}^{\infty} \gamma^k f(k) = \sum_{n=0}^{\infty} 2^n \sum_{k=1}^{\infty} \gamma^k [\![ f(k) = 2^n ]\!]$$

$$= \sum_{n=0}^{\infty} 2^n \sum_{c=0}^{\infty} \gamma^{(2c+1)2^n} \quad \text{(Lemma 10)}$$

$$= \sum_{n=0}^{\infty} 2^n \gamma^{2^n} \sum_{c=0}^{\infty} \left(\gamma^{2^{n+1}}\right)^c = \sum_{n=0}^{\infty} 2^n \gamma^{2^n} \frac{1}{1-\gamma^{2^{n+1}}}$$

$$\leq \sum_{n=0}^{\infty} 2^n \gamma^{2^n} \left(1 + \frac{\gamma}{2^{n+1}(1-\gamma)}\right) \quad \text{(Lemma 14)}$$

$$= \sum_{n=0}^{\infty} 2^n \gamma^{2^n} + \frac{1}{2}\frac{\gamma}{1-\gamma} \sum_{n=0}^{\infty} \gamma^{2^n}$$

$$\leq \frac{1}{1-\gamma}\left(\frac{1}{e} + \frac{\gamma}{\ln 2} + \frac{\gamma}{2} + \frac{\gamma^2}{2}\left(\log_2 \ln 4 + \log_2 \frac{1}{1-\gamma}\right)\right)$$

$$\leq \frac{1}{1-\gamma}\left(\frac{1}{e} + \frac{1}{\ln 2} + \frac{1}{2}\log_2 \ln 16 + \frac{1}{2}\log_2 \frac{1}{1-\gamma}\right)$$

where we used Lemma 11 and Lemma 12 on the second to last line together with $\ln \frac{1}{\gamma} \geq 1 - \gamma$. $\quad\square$

**Lemma 14.** For $\gamma \in [0, 1)$ and $a \geq 1$:

$$\frac{1}{1-\gamma^a} \leq 1 + \frac{1}{a}\frac{\gamma}{1-\gamma} \,.$$

*Proof.* For $\epsilon > 0$ and $a \geq 1$, it can be shown that $(1 + \varepsilon)^a \geq 1 + a\varepsilon$. Then, taking $\gamma := \frac{1}{1+\varepsilon}$:

$$
\begin{aligned}
(1 + \varepsilon)^a \geq 1 + a\varepsilon \quad &\Leftrightarrow \quad (1 + \varepsilon)^a - 1 \geq a((1 + \varepsilon) - 1) \\
&\Leftrightarrow \quad \frac{1}{(1 + \varepsilon)^a - 1} \leq \frac{1}{a((1 + \varepsilon) - 1)} \\
&\Leftrightarrow \quad \frac{1}{\gamma^{-a} - 1} \leq \frac{1}{a(\gamma^{-1} - 1)} \\
&\Leftrightarrow \quad \frac{\gamma^a}{1 - \gamma^a} \leq \frac{\gamma}{a(1 - \gamma)} \\
&\Leftrightarrow \quad \frac{1}{1 - \gamma^a} \leq 1 + \frac{1}{a}\frac{\gamma}{1 - \gamma} \,,
\end{aligned}
$$

which proves the result. □

## Footnotes

[5] https://oeis.org/A006519.