[Reviews · NeurIPS 2018]

Reviewer 1



Summary: This paper considers the problem of tree-based search by leveraging a given (learned or hand-designed) policy to solve goal-based search problems. The paper proposes two different search algorithms with varying strengths and weakness, and provides guarantees on the number of nodes to be expanded before reaching a goal state. First, LevineTS based on Levine Search performs best-first search enumeration: penalizes with depth of a node and performs state cuts using the policy. Second, LubyTS is based on existing work on scheduling for randomized algorithms. It takes advantage of large number of candidate solutions (goal states) in the form of known maximum depth at which these solutions are present. It samples trajectories of this length using the policy until solution is found. Experiments are performed on Sokoban domain using a neural network policy learned using a reinforcement learning algorithm and comparison is performed with a domain-independent planner with good results. Pros: - Very well-written paper. - The proposed tree search algorithms and guarantees on expected search time (in terms of expanded nodes) to reach goal state. - Technically solid work. Cons: - The practical significance of this problem and tree search algorithms is not clear. - Lack of experimental results on more practical real-world applications of this problem setting and algorithms. Detailed Comments: 1. Section 2 introduces too much notation that I don't think is necessary to explain the problem setting and algorithms. These are very simple search concepts. 2. I would have at least liked to see some potential real-world use-cases of the problem setting considering this paper is submitted to a NIPS conference. 3. Experimental results validate the algorithms and guarantees, but it would have been nice to see more empirical analysis on real-world use cases.

Reviewer 2



The paper presents two planning algorithms based on tree search. The novelty of these algorithms is the use of a policy based criterion instead of a standard heuristic to guide the search. The first algorithm (LevinTS) uses a cost function d(n)/\pi(n) which ensures nodes are expanded in a best-first order. This allows the authors to derive an upper bound to the number of expansions performed before to reach a target node. The second algorithm (LubyTS) is a randomized algorithm based on the sampling of trajectories. Using a specific restarting strategy, the authors derived an upper-bound to the expected number of expansions before reaching a target state. Both the algorithms are analyzed under the assumption of a deterministic environment. Comments ------------ The paper is overall well written and easy to follow for the part concerning the algorithms. What is less clear to me is the context and motivations. - This paper is about planning given a baseline policy. The main issue I see here is that if the provided policy has zero or small probability of reaching the goal, the algorithms will fail. In the case of non-zero (small) probability, LevinTS is still able to find an optimal solution but with a very large number of node expansions (see Thm 3). I suppose that this is the reason why you pre-trained the policy using RL in the experiments (this allows overcoming the issue). This is a strong weakness of the proposed approach. Model-based algorithms (LevinTS is model-based) for planning do not have such requirements. On the other hand, if the goal is to refine a policy at the end of some optimization procedure I understand the choice of using a policy-guided heuristic. - Concerning LubyTS it is hard to quantify the meaning of the bound in Thm. 6 (the easy part is to see when it fails, as mentioned before). There are other approaches based on generative models with guarantees, e.g., (Grill, Valko, Munos. Blazing the trails before beating the path: Sample-efficient Monte-Carlo planning, NIPS 2016). How do you perform compared to them? - Your setting is very specific: you need to know the model or/and have access to a generative model (for expanding or generating trajectories), the problem should be episodic and the reward should be given just at the end of a task (i.e., reaching the target goal). Can you extend this approach to more general settings? - In this paper, you perform planning offline since you use a model-based approach (e.g., generative model). Is it possible to remove the assumption of the knowledge of the model? In that case, you would have to interact with the environment trying to minimize the number of times a "bad" state is visited. Since the expansion of a node can be seen as an exploratory step, this approach seems to be related to the exploration-exploitation dilemma. Bounding the number of expansions does not correspond to having small regret in general settings. Can you integrate a concept related to the long-term performance in the search strategy? This is what is often done in MCTS. All the proposed approaches have weaknesses that are partially acknowledged by the authors. A few points in the paper can be discussed in more details and clarified but I believe it is a nice contribution overall. -------- after feedback I thank you for the feedback. In particular, I appreciated the way you addressed the zero probability issue. I think that this is a relevant aspect of the proposed approaches and should be addressed in the paper. Another topic should be added to the paper is the comparison with Trailblazer and other MCTS algorithms. Despite that, I personally believe that the main limitation is the assumption of deterministic transitions. While the approach can be extended to stochastic (known) models I would like to know how the theoretical results become.

Reviewer 3



This paper considers the problem of searching for an action sequence leading to goal states in a deterministic MDP. Two tree search algorithms, obtained by incorporating a stochastic policy (constructed using reinforcement learning in the paper) as guidance in the Levin search and Luby search, are presented, with complexity analysis. The algorithms are evaluated on 1,000 computer generated levels of Sokoban. Comments: - Line 56: T(n) not been defined yet. - LevinTS is obtained from Levin search by taking 'computation time' as 'depth of a node', and LubyTS seems to be a direct application of Luby search. It'll be helpful to describe what the novelty is, particularly on the analysis, which is a major part of the paper. (I'm not familiar with Levin search and Luby search) - In the experiments, LubyTS does not perform well as compared to LevinTS and LAMA. For LevinTS, it is not clear whether the stochastic policy obtained using reinforcement learning is helpful. In fact, LevinTS using a noisy version of the stochastic policy performs slightly better, and suggests a need to compare with LevinTS using uniform stochastic policy. *** After rebuttal The paper contains interesting ideas, but I still have some concern on the effectiveness of the proposed approach because the experimental evaluation was limited to one problem and had mixed performance (LubyTS does not perform well, and LevinTS traded off time for shorter solutions as compared to LAMA). That said, I won't be upset if the paper is accepted.